# CAUSAL GRAPH RECOVERY IN NEUROIMAGING THROUGH ANSWER SET PROGRAMMING

## ABSTRACT

Learning directed causal graphs from time-series data poses significant challenges, especially in fMRI where slow sampling rate obscures fast neural interactions. This temporal mismatch leads to undersampling, which can make multiple graphs equally plausible. We address this problem by explicitly modeling undersampling effects when recovering causal graphs. Our approach employs answer set programming (ASP) to enforce domain-specific constraints and optimize soft observational constraints, thereby identifying a Markov equivalence class for the resulting graph solutions. By customizing an ASP solver to collect multiple near-optimal solutions, we obtain not only the single best-fitting graph but an equivalence class of high-scoring graphs for expert consideration. This method, called Real-world noisy RASL (RnR), can also act as a meta-solver: it refines the output of other causal discovery algorithms by accounting for undersampling biases. In simulations and empirical brain network data, RnR produces more accurate causal graphs than state-of-the-art methods, improving F1-scores by an average of 12% by reducing false connections. We demonstrate that RnR is robust to varying undersampling rates – maintaining high precision and recall even as sampling becomes sparser – whereas competing methods degrade significantly. Finally we test RnR on open-ended questions without know ground truth like human brain fRMI data, showing that incorporating undersampling-aware constraints via ASP yields more reliable and interpretable brain connectivity estimates from fMRI time series, bridging the gap between neural dynamics and observational data.

## 1 INTRODUCTION

Causal inference from functional Magnetic Resonance Imaging (fMRI) data has emerged as a critical endeavor to understand the neural mechanisms underlying cognitive processes and behaviors. Researchers not only seek to identify active brain regions during tasks, but also unravel the causal relationships between these regions, often referred to as "effective connectivity" (Friston, 1994). Graphical causal models, such as causal Bayesian networks, have become a popular framework for this purpose, combining directed graphs with joint probability distributions to model the dependencies between different brain regions (Pearl, 2009). These models adhere to the Causal Markov Condition, which asserts that each node in a causal graph is conditionally independent of its non-descendants given its parents (Spirtes et al., 2001).

However, applying these models to fMRI data is fraught with challenges, particularly due to the mismatch between the temporal resolution of fMRI and the rapid timescale of neural processes. Typical sampling intervals in fMRI, ranging from one to three seconds, are much slower than the millisecond-level interactions between neurons, leading to significant undersampling (Valdes-Sosa et al., 2011). This undersampling often results in multiple causal graphs being statistically indistinguishable given observed data, forming what is known as a Markov Equivalence Class (Pearl, 2009; Spirtes et al., 2001). These problems are exacerbated by the indirect nature of the *Blood-Oxygen-Level Dependent* (BOLD)[1] signal, which reflects neural activity through complex and variable hemodynamic responses (Handwerker et al., 2004).

---

[1]BOLD signal is the measurement obtained from fMRI that reflects changes in blood oxygenation related to neural activity. When neurons fire, blood flow to that region changes; fMRI captures these changes at a slow timescale (on the order of 1–2 seconds per sample). Importantly, the BOLD signal is an indirect and delayed proxy of neural firing.

In addition, the inherent variability in the hemodynamic response across different brain regions and subjects, adds another layer of complexity. Variations in the time-to-peak of the BOLD response can lead to incorrect inferences about the direction of causality, particularly when using methods like Granger causality, which assumes a fixed temporal relationship between cause and effect (David et al., 2008; Seth et al., 2013). Although some studies suggest that Granger causality may be robust to certain variations in the hemodynamic response (Seth et al., 2013), the combination of measurement noise, undersampling, and hemodynamic variability often undermines the reliability of causal inferences from fMRI data.

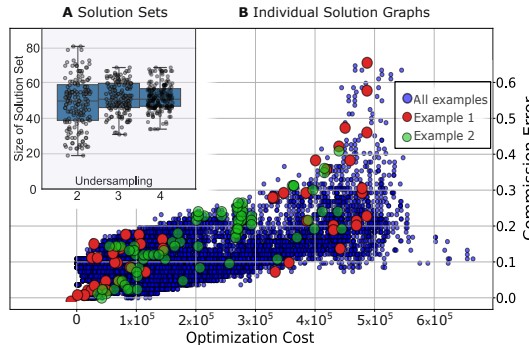

Figure 1: Top left(A): Size of optimization solution set across different undersamplings, repeated 100 times. Bottom right(B): Commission error of the solution vs. optimization cost for that solution. Solutions in one equivalent class are highlighted in red.

In response to these challenges, this paper proposes a novel approach called **RnR** that explicitly accounts for the effects of undersampling in the derivation of causal graphs. By employing constraint optimization through Answer Set Programming (ASP), we aim to identify the most probable causal graph from a set of potential candidates. ASP allows for the incorporation of domain-specific knowledge and constraints, facilitating the identification of not only a single graph but an equivalence class of possible graphs, thereby offering a more comprehensive understanding of the underlying causal structure (Gebser et al., 2012). We validate our approach using both simulated data and real fMRI data, demonstrating its superiority over existing methods . Our results suggest that ASP offers a powerful new tool for causal inference in neuroimaging, providing more accurate and intuitive insights into the brain's functional architecture.

## 2 BACKGROUND

We study recovery of the true causal graph $\mathcal{G}^1$ from an observed, undersampled measurement graph $\mathcal{H}$ when the sampling rate $u$ is unknown. We first fix notation and core facts, then state the limitation that motivates our approach.

A directed dynamic causal model extends standard causal models (Pearl et al., 2000; Spirtes et al., 1993) by adding time: variables $V_{1:n}$ appear at times $t, t-1, \ldots$, and a first-order Markov structure is assumed so that $\mathbf{V}^t \perp\!\!\!\perp \mathbf{V}^{t-k} \mid \mathbf{V}^{t-1}$ for $k > 1$ (Spirtes et al., 2000). When sampling is slow relative to neural dynamics, fMRI produces an undersampled graph $\mathcal{G}^u$ that can differ from $\mathcal{G}^1$ (Danks & Plis, 2013; Gong et al., 2015). In the compressed representation, an edge $i \to j$ in $\mathcal{G}^u$ occurs iff there is a directed path of length $u$ from $i$ to $j$ in $\mathcal{G}^1$; a bidirected edge $i \leftrightarrow j$ occurs if a common ancestor reaches both with equal path length $< u$ (Danks & Plis, 2013). This view clarifies which dependencies are artifacts of rate.

To address the structural implications of undersampling, several approaches have been developed. For instance, Danks & Plis (2013) explored the problem structurally, while Gong et al. (2015) provided a parametric approach for two-variable systems. This line of work has been extended to more general subsampled and temporally aggregated VAR processes, including constraint-optimization methods for subsampled time series data (Hyttinen et al., 2017), causal discovery from temporally aggregated time series (Gong et al., 2017), and recent proxy-variable approaches that can in principle identify the full causal graph from subsampled time series without parametric assumptions (Liu et al., 2023).

Dealing with latent confounders, such as those introduced by undersampling, has led to the development of various graph representations, including Partially-Observed Ancestral Graphs (PAGs) (Zhang, 2008) and Maximal Ancestral Graphs (MAGs) (Richardson & Spirtes, 2002). However, these frameworks often struggle with the complexities introduced by undersampling (Mooij & Claassen, 2020). Compressed graphs are effective for undersampling: they are 1–1 equivalent to

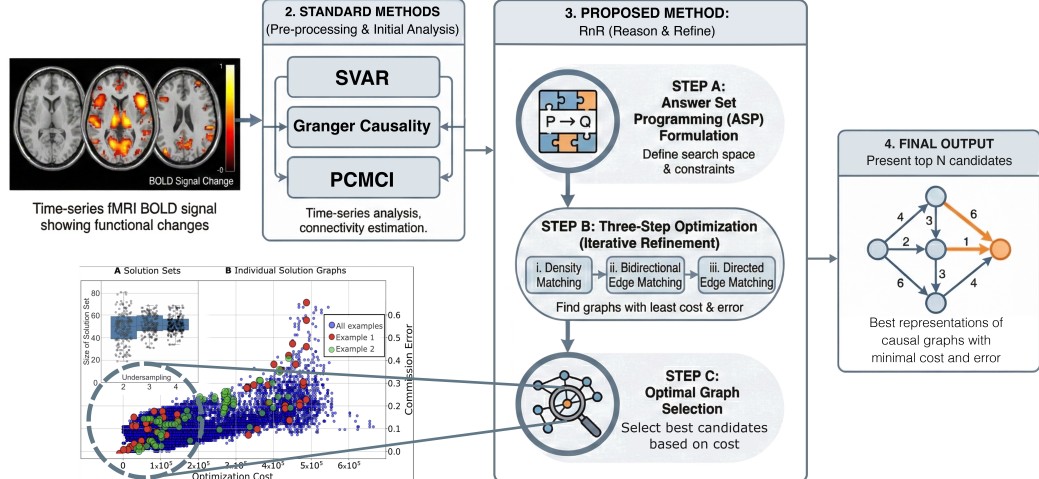

Figure 2: Overview of our pipeline. Starting from 4D fMRI volumes, classical methods extract a single "measured" graph from BOLD time series (e.g., using Granger Causality, SVAR, or PCMCI). However, these approaches typically ignore the effects of temporal undersampling. Our method extends beyond this step by accounting for the ambiguity introduced by slow sampling. Instead of selecting a single best-fit graph, we recover an equivalence class of plausible causal graphs that are consistent with both the observed data and the structural distortions caused by undersampling. Finally we present top $N$ graphs based on optimization cost.

dynamic ADMGs and allow computing rate effects by path length, and they outperform PAG/MAG-style formulations under unknown $u$ (Plis et al., 2015; Abavisani et al., 2023).

The Rate-Agnostic Structure Learning (RASL) algorithm (Plis et al., 2015) addressed the challenge of causal inference from undersampled data by directly tackling undersampling while adopting a rate-agnostic approach that avoids assumptions about the undersampling rate. RASL was a big leap in computational efficiency. It cleverly checked all possible rates and used "stopping rules" to keep from re-exploring the same parts of a graph. Its main drawback, however, was that it didn't scale well to larger graphs, leaving room for improvement. Building on this foundation, newer approaches to causal learning have been developed that are far more scalable and efficient.

Building on these foundations, recent advancements in causal structure learning have led to the development of generalized rate-agnostic approaches that significantly enhance the scalability and efficiency of these methods. The reformulation of RASL into a constraint satisfaction framework, expressed using a declarative language, has shown considerable promise by enabling more efficient analysis of large graphs called Solver-based RASL (sRASL) (Abavisani et al., 2023). By incorporating constraints from strongly connected components (SCCs)[2] this approach offers a scalable and accurate method for causal inference from undersampled data. It can analyze graphs with over 100 nodes—a significant improvement on earlier methods that struggled with much smaller graphs. The sRASL method improves solving time 1000-fold while maintaining the same theoretical guarantees as its predecessors. In parallel, Solovyeva et al. (2023) showed that combining measurement graphs at multiple, deliberately undersampled rates can further shrink the equivalence class of system-timescale graphs and introduced an ASP-based solver (dRASL) that generalizes the RASL/sRASL encoding to multiple measurement timescales. However, sRASL shows limitations on real-world datasets, as the performance gains seen in simulations do not fully translate to noisy, practical conditions.

In Sanchez-Romero et al. (2019b), several methods were tested on synthetic BOLD data with feedback (e.g., Granger regression, MVAR, FASK, Two-Step). Many reached over 80% orientation precision and recall, including for 2-cycles. That study, however, did not consider that BOLD is

---

[2]Subsets of nodes in a directed graph where each node is reachable from every other node in the subset. In other words, an SCC is a "maximal loop" or cluster of nodes with mutual reachability. In our method, we allow nodes within an SCC to be interlinked (modeling instantaneous or cyclic relationships), but we enforce that the graph of SCCs (each SCC treated as a single super-node) has no directed cycles, making it a DAG.

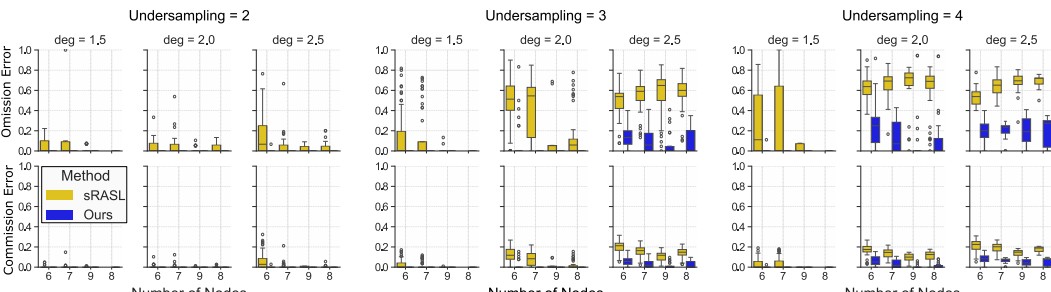

Figure 3: Normalized omission (top) and commission (bottom) errors for edge-breaking experiments with varying undersampling rates and graph densities, comparing the original approach with our improved sRASL-based method.

undersampled. Here we revisit those algorithms under sparse sampling and compare them with an ASP-based approach (RnR).

The method space has grown (e.g., LiNGAM (Shimizu et al., 2006), EC-GAN (Kim et al., 2021), amortized transformers (Paul et al., 2022), and RL-style models (Mnih et al., 2016; Salehi et al., 2021; Pamfil et al., 2020)). Outside fMRI, related challenges arise when time series are irregularly sampled or acquired below the Nyquist rate. Bahadori & Liu (2012) and Heerah et al. (2021) adapt Granger-causality analysis to irregularly sampled physiological and biological time series, providing tests that explicitly account for unequal sampling intervals rather than forcing data onto a regular grid. In genomics, CaSPIAN and related "causal compressive sensing" approaches combine compressed sensing with Granger causality to infer directed gene–gene interactions from heavily undersampled expression data (Emad & Milenkovic, 2014). Kathpalia & Nagaraj (2023) further show that, under suitable structured sensing matrices, Granger-causal relations can be recovered directly from compressively sensed sparse signals, enabling causal analysis even when acquisition is far below Nyquist. RnR fits into this broader methodological landscape by focusing on undersampling and hemodynamic distortion in fMRI, but using an ASP-based encoding that is, in principle, applicable to other subsampled or irregularly sampled modalities.

For clean comparison and interpretability, we benchmark the widely used set from Sanchez-Romero et al. (2019b): GIMME (Gates & Molenaar, 2010), MVGC (Barnett & Seth, 2009), MVAR (Bressler & Seth, 2003), and FASK (Sanchez-Romero et al., 2019a). These cover group and individual modeling, multivariate dependencies, and non-Gaussian orientation cues.

We then evaluate all methods on undersampled data and against ASP. This isolates robustness to undersampling. Among baselines, FASK performs well; RnR further improves accuracy by making undersampling explicit. Detailed results appear in Section 4.2.

## 3 METHODS

In this section, we present Real-world noisy RASL (RnR)[3], our enhanced structure learning framework, tailored for fMRI connectivity analysis. Our method, RnR, builds upon the sRASL framework through several key innovations tailored for noisy, undersampled fMRI data. These contributions include: (1) utilizing custom made Answer Set Programming solver to retrieve all plausible, near-optimal graphs within a cost threshold instead of a single solution; (2) constraining the optimization with realistic edge density limits to ensure the resulting graphs are biologically plausible; (3) acting as a meta-solver to refine the outputs of other causal discovery algorithms by systematically handling undersampling effects; (4) employing a prioritized, multi-stage process that sequentially matches graph density, resolves links, and orients edges; and (5) applying an adaptive weighting

---

[3]A note on naming: "RASL" stands for Rate-Agnostic Structure Learning, an approach introduced by Plis et al. (2015) for causal discovery without knowing the true sampling rate. sRASL for structural RASL used ASP to find a single optimal causal graph consistent with undersampled data Abavisani et al. (2023). Our RnR extends this by considering real-world noise (hence "Real-world noisy RASL") and retrieving multiple solutions.

scheme to the cost function to discourage spurious connections while retaining high-confidence ones.

### 3.1 OVERVIEW OF THE SRASL FRAMEWORK

The original sRASL framework (Abavisani et al., 2023) is an ASP-based approach that takes an initial directed graph $\mathcal{H}$ and applies structural constraints to refine it. We follow Abavisani et al. (2023) and use Clingo[4] for our implementation. sRASL allows cycles within strongly connected components (SCCs), but requires the SCCs themselves to form a Directed Acyclic Graph (DAG). This DAG-over-SCCs structure prevents latent-level causal cycles. The original sRASL only returned a single optimal solution minimizing a cost function, but this posed limitations. Due to inherent uncertainty in the case of undersampling, the solution with the least cost is not necessarily the correct answer. Therefore, ignoring near-optimal alternatives in noisy data can be misleading.

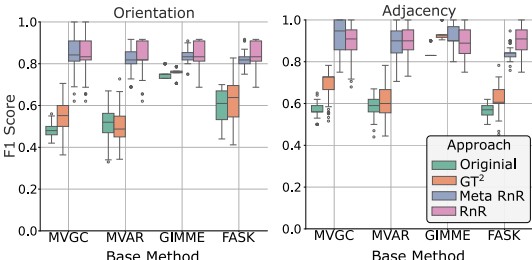

Figure 4: Performance comparison using Sanchez-Romero's data with and without the RASL meta-solver. Applying RASL on top of traditional methods improves accuracy by accounting for undersampling effects, with additional gains achieved by using PCMCI.

### 3.2 DENSITY CONSTRAINT FOR REALISTIC GRAPHS

Neuroscience knowledge can inform the expected density of functional brain networks. Empirical studies of brain connectivity (both functional and structural) show that brain graphs tend to be neither nearly empty nor fully connected; they often have a moderate density (for example, on the order of 10–30% of all possible edges, depending on node definition (Smith et al., 2011). After running sRASL, we observed that some solutions were unrealistically sparse or dense, likely because the optimization only cared about matching $\mathcal{H}$ and not an absolute scale of connectivity. To correct this, we introduce an explicit density constraint in our ASP formulation:

```
1  countedge1(C):- C = #count { edge1(X, Y): edge1(X, Y), node(X),
       node(Y)}.
2  countfull(C):- C = n*n.
3  hypoth_density(D) :- D = 1000*X/Y,  countfull(Y), countedge1(X).
4  abs_diff(Diff) :- hypoth_density(D), Diff = |D - d|.
5  :~ abs_diff(Diff). [Diff@2]
```

Line $1 - 3$ calculates graph density in the present hypothetical solution, and lines $4, 5$ measure the difference from ground truth and penalize when value `Diff`$> 0$. This prevents trivial empty or fully connected graphs and ensures structural realism based on known brain network densities (which are reasonably well-estimated and biologically bound).

### 3.3 OPTIMIZATION FOR MULTIPLE SOLUTIONS PRIORITIZED BY CONNECTION TYPE

To address the limitation of single-solution output, we modified Clingo to return all graphs $G$ satisfying $C(G) \leq \Delta$, where $C(G)$ is the cost of graph $G$ and $\Delta$ is a tolerance percentage relative to graph size. This ensures the method scales naturally with graph size, as a fixed integer tolerance would be overly restrictive for large graphs and too loose for small ones. Through hyperparameter tuning, we found out that $1.9$ times the cost of the solution with minimum cost in solution set the best threshold for $\Delta$ (Refer to Appendix B for details).

To efficiently recover causal graphs under severe noise and undersampling, we adopt a prioritized multi-stage optimization strategy that decomposes the total cost into structurally meaningful components. Our aim is to minimize the overall cost $C(G)$. We decompose the total cost as:

---

[4]Clingo is an open-source solver for Answer Set Programming (ASP) that finds solutions (called "answer sets") to a given set of logical rules and constraints. In our context, Clingo explores different possible graphs and identifies those that satisfy all our constraints while optimizing a cost function.

$$C(G) = C_o(G) + C_b(G) + C_d(G), \tag{1}$$

corresponding respectively to penalties on density (edge count), bidirected structure, and edge orientation. More specifically, we use the following stepwise cost minimizations:

$$\min_{G \in \mathcal{G}} \quad C_o(G) = \lambda_o \cdot |\text{density}(G) - \text{density}(G^*)| \tag{2}$$

$$\text{Feasible set } \mathcal{F}_1 : \quad \mathcal{F}_1 = \{G \in \mathcal{G} \mid C_o(G) \leq \epsilon_1\}$$

$$\min_{G \in \mathcal{F}_1} \quad C_b(G) = \lambda_b \cdot \# \{\text{bidirected mismatch to } G^*\} \tag{3}$$

$$\text{Feasible set } \mathcal{F}_2 : \quad \mathcal{F}_2 = \{G \in \mathcal{F}_1 \mid C_b(G) \leq \epsilon_2\}$$

$$\min_{G \in \mathcal{F}_1, \mathcal{F}_2} \quad C_d(G) = \lambda_d \cdot \# \{\text{directed mismatch to } G^*\} \tag{4}$$

where $G$ denote a candidate causal graph, $\mathcal{G}$ the space of all possible graphs, $G^*$ the ground-truth. $\epsilon_1$ and $\epsilon_2$ are tolerance parameters. Priority weights $\lambda_d \gg \lambda_b \gg \lambda_o$ reflect the order of importance. Unlike continuous hyperparameters in gradient-based learning, the weights $\lambda_o, \lambda_b, \lambda_d$ in our framework function as discrete hierarchical optimization levels. We determined the optimal hierarchy through hypothesis testing (for detailed analysis of hyperparameter tuning refer to Appendix A). This approach constrains the search space in a lexicographic manner: first matching global sparsity, then resolving stable structural backbones, and finally fine-tuning edge directions. For a more detailed discussion about optimization and algorithmic treatment of proposed RnR, please refer to Appendix C

### 3.4 RnR as a Meta-Solver for Undersampling Effects

A significant challenge in causal learning from fMRI is handling unobserved common causes (latent confounders) that arise from undersampling. For example, if two brain regions $A$ and $B$ are both driven by an unmeasured region or stimulus (common cause) that was not captured due to slow sampling, standard algorithms might either link $A-B$ with directed edges in a loop or miss the connection altogether. Bidirected edges (often denoted $A \leftrightarrow B$) can represent unresolved causality due to a hidden common cause in causal graphs. Some time-series causal discovery methods (e.g., Granger causality (Granger, 1969), VAR models (Lütkepohl, 2005), and the PCMCI algorithm (Runge et al., 2019)) are capable of producing bidirected or contemporaneous links. However, many algorithms that ignore undersampling (e.g., standard PC or GES variants, Granger causality in its basic form, etc.) will not output bidirected edges at all, potentially misrepresenting the true connectivity.

We address this issue by using RnR as a meta-solver: RnR takes as input any initial graph $\mathcal{H}$ produced by a first-order method, but then refines it. If the initial method did not account for undersampling, $\mathcal{H}$ might contain spurious patterns (e.g., a two-node cycle $A \rightarrow B$ and $B \rightarrow A$) or be missing bidirectional links. We therefore augment $\mathcal{H}$ before running ASP: for any pair of nodes that form a mutual directed cycle ($A \rightarrow B$ and $A \leftarrow B$) in $\mathcal{H}$, we interpret this as evidence of a possible latent confounder. We then modify $\mathcal{H}$ by adding a bidirected edge $A \leftrightarrow B$. Additionally, to encode uncertainty about direction of influence, we include both directed edges $A \rightarrow B$ and $A \leftarrow B$ in $\mathcal{H}$ (if not already present), but assign them a low weight $w^-$ (penalty) for inclusion.

The ASP solver thus starts with an enriched graph containing a bidirectional link and two low-confidence directed links between $A$ and $B$. The solver can decide, through optimization, whether the final solution should keep the bidirected connection, one of the directed connections, or both, depending on what best satisfies all constraints and minimizes cost. Essentially, we are giving the solver the flexibility to explain an observed correlation between $A$ and $B$ as a latent confounding (bidirected edge), as a direct causal relationship (one of the directed edges), or as a mix of both, whichever is most consistent with the data and constraints. By using RnR as a post-processor in this way, we "inject" undersampling awareness into any algorithm's output. In Section 4.2, we demonstrate this meta-solver capability by applying RnR to outputs from the method of Sanchez-Romero et al. (2019a) and from PCMCI, yielding improved results.

### 3.5 IMPLEMENTATION[5] AND PRACTICAL CONSIDERATIONS

**Computational Efficiency:** Running Clingo with optN on our problem sizes (graphs of up to 10 nodes in our simulations) was fast (under a few seconds) for each instance. For larger graphs (e.g., 20-50 nodes), ASP solving could become slower. However, our modular constraints (SCC-based DAG) significantly reduce complexity by disallowing many cyclic possibilities upfront. Further optimization or heuristic solving is a direction for future work, but for the scope of our experiments, the approach was tractable.

**Validation of Multiple Solutions:** Importantly, retrieving multiple solutions allowed us to examine the stability of the recovered graph. We found that often many of the top solutions shared most edges in common, differing only in a few uncertain connections. This gave us confidence that our method identifies a core reliable structure plus a small set of alternative edges. We illustrate this in Figure 1A, which plots the number of solutions vs. undersampling rate: even at high undersampling, the solution set remains of modest size (tens of graphs), and in Figure 1B we show the relationship between solution cost and edge errors. Notably, we observed a positive correlation between optimization cost and edge commission error (false positive rate) across solutions. However, some higher-cost solutions had low commission error, indicating that near-optimal graphs can sometimes be almost as accurate as the optimal one. This justifies examining a small pool of top solutions. In practice, we found that the best solution (by ground-truth error) was often among the top 10 by cost. Therefore, our strategy is to take the top-$N$ solutions from Clingo (we used $N = 10$) and then select the final graph by an additional criterion (e.g., best fit on a validation dataset or expert judgment). This approach yielded robust identification of the true causal graph in our tests.

## 4 RESULTS

### 4.1 IMPROVED EDGE-BREAKING EXPERIMENT

We replicated and extended the edge-breaking experiment from Abavisani et al. (2023) to demonstrate the robustness of our approach on graphs with intentionally broken edges. In this experiment, we generated causal ground truth graphs $\mathcal{G}^1$ and undersampled them at various rates, then simulated noise by randomly deleting an edge. The goal was to test the ability of our method to recover the true graph structure, compared to the original sRASL. The results, depicted in Figure 3, show that our method consistently achieved lower omission and commission errors[6] compared to the original approach, even under high undersampling conditions. This illustrates that our improved algorithm can more effectively recover the true graph structure, including robustness against edge-breaking perturbations.

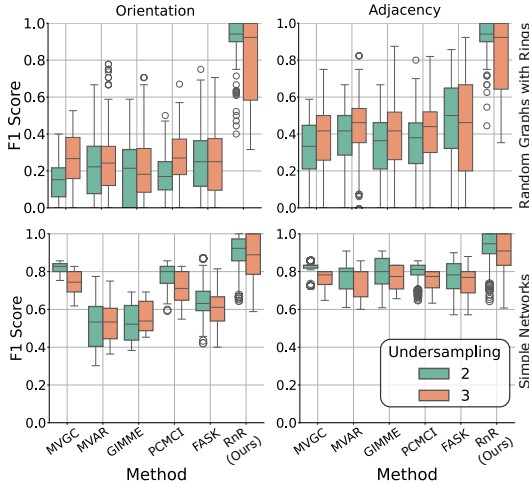

Figure 5: Comparison of Sanchez-Romero's simple data generation approach with larger, more complex VAR-generated graphs. Sanchez-Romero's data lacks real loops and is limited in scope, which may affect the generalizability of causal inference results.

### 4.2 BOLD SIMULATION DATA: ENHANCING CAUSAL INFERENCE WITH RASL AS A META-SOLVER

We further evaluated RnR on simulated data from Sanchez-Romero et al. (2019b). These data have been widely used within the neuroscience community, but were generated with small, sim-

---

[5]For a brief introduction to Clingo and Answer Set Programming, please refer to Appendix D

[6]A commission error is a false positive (i.e., incorrectly added edge). An omission error is a false negative (i.e., incorrectly omitted edge). We report these separately to characterize different types of errors (adding spurious links vs. missing real links).

ple graphs lacking real loops and using an idiosyncratic data generation method. We demonstrate that the methods that they study, including ones developed for these data (e.g., (Sanchez-Romero et al., 2019a)), do not account for the effects of undersampling, which can bias causal inference.

Our approach applied RnR as a meta-solver: after running Sanchez-Romero's original methods, we applied RnR to the resulting graph to explicitly incorporate the effects of undersampling. This adjustment led to improved accuracy, as RnR optimized the causal graph structure to better reflect the underlying dynamics. In essence, RnR served as an enhancement layer, correcting for undersampling effects ignored by previous methods (Figure 4).

Additionally, we explored the use of PCMCI as an alternative to standard methods like SVAR and Granger Causality (GC). Cook et al. (2017) had previously demonstrated that SVAR and GC perform well for scenarios with isochronal bidirected edges, which arise due to undersampling, as well as directed edges. However, we observed that PCMCI, initially introduced by Moneta et al. and later significantly improved by Runge, performs better in these scenarios. Therefore, we incorporated PCMCI with RASL, achieving more accurate results by effectively handling undersampling and improving causal inference. Figure 4 illustrates these improvements.

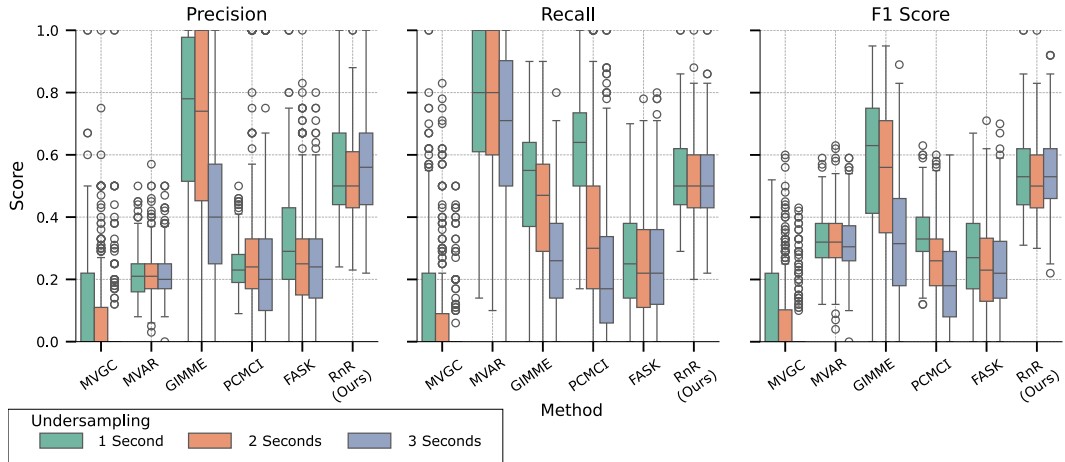

Figure 6: Impact of different undersampling rates (1s, 2s, 3s) on BOLD signal preservation in fMRI data simulated with the balloon model. Minimal error is observed with 1-second undersampling, whereas larger intervals degrade accuracy in all methods that don't account for undersampling effect. Our method RnR accounts for this effect and does not suffer loss from undersampling.

### 4.3 ANALYZING THE LIMITATIONS OF SANCHEZ-ROMERO'S DATA GENERATION

Despite its widespread acceptance, Sanchez-Romero's data generation approach is limited in scope and complexity. It employs simple, small graphs without real loops and is generated using a specific, contrived process. To demonstrate the limitations of this dataset for broader causal inference applications, we conducted experiments with larger VAR models on graphs with many variables and more complex structures, showcasing the limitations of Sanchez-Romero's setup (Figure 5). This underscores the importance of considering more realistic and complex data when testing causal inference methods.

### 4.4 IMPACT OF UNDERSAMPLING ON BOLD SIGNAL IN FMRI DATA

We also examined the effect of undersampling on time series data generated from a VAR model and then processed through a balloon model to simulate the BOLD response in fMRI data (Buxton et al., 1998). Given the smooth nature of the BOLD signal, undersampling by one-second intervals often leads to minimal information loss, while undersampling by larger intervals (e.g., two or three seconds) introduces significant inaccuracies. Figure 6 illustrates how different undersampling rates impact the preservation of temporal information in BOLD data, underscoring that aggressive undersampling can obscure meaningful connectivity patterns.

### 4.5 COMPARATIVE ANALYSIS OF CAUSAL DISCOVERY METHODS ON fMRI DATA

To validate our causal discovery framework and demonstrate its ability to extract novel information from neural imaging data, we conducted a comparative analysis using Granger Causality Mapping (GCM) Roebroeck et al. (2005) and our RnR. We utilized the same six ICA-derived brain regions as Sridharan et al. (2008), spanning the Central Executive Network (CEN), Salience Network (SN), and Default Mode Network (DMN). By paying close attention to causal rates and sampling distinct causal properties, our results reveal that different methods can uncover complementary layers of network organization.

We first applied GCM to validate our pipeline against established findings. Our in-house implementation (Figure 7 left) showed substantial agreement with Sridharan et al. (2008)'s results. Specifically, we confirmed the central role of the right fronto-insular cortex (rFIC) in mediating network interactions. As shown in Figure 7 left and Table 1, the strongest connection was the coupling between rFIC and the right dorsolateral prefrontal cortex (rDLPFC), replicating the key salience-executive link emphasized in the original study. Furthermore, we replicated the finding that the Salience Network exerts influence over the DMN, with robust directed connections from rFIC and ACC to DMN regions (VMPFC and PCC). These agreements validate the fidelity of our implementation and the quality of the underlying data processing.

Applying our RnR method to the exact same ICA components yielded a different but related set of answers, highlighting the method's sensitivity to structural rather than purely temporal causal signals. While GCM emphasized bidirectional coupling (rFIC ↔ rDLPFC), RnR identified a clear hierarchical structure driven by the Default Mode Network (Figure 7 right). The most significant finding was a robust unidirectional connection from VMPFC to rFIC (Figure 7 right, Table 1), which was present in nearly 90% of solutions—a far stronger signal than any individual connection found by GCM.

Comparing the two approaches reveals important convergences and one critical divergence. **Similarities:** Both methods agree on the high level of interconnectivity between the three networks and the central importance of the Salience Network (rFIC/ACC) in bridging distinct functional systems. **Key Difference:** The methods disagree on the primary driver of the system. GCM identifies the **reciprocal temporal dynamics** between rFIC and rDLPFC as the dominant feature, likely reflecting the active "switching" mechanism. In contrast, RnR identifies the **structural precedence** of VMPFC over rFIC, suggesting that the DMN's state may structurally constrain or gate the activity of the Salience Network. This distinction hints that by considering undersampling effect, we can see a bigger picture in brain causal structure discovery.

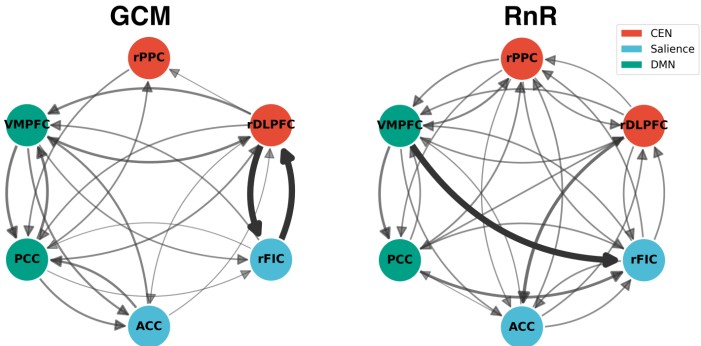

Figure 7: Comparative Analysis of (GCM) and (RnR). **Left** GCM network showing strong bidirectional coupling between rFIC and rDLPFC (thick black edges), replicating Sridharan et al. (2008)'s emphasis on salience-executive interactions. **Right** RnR network reveals a distinct structural hierarchy. Unlike the bidirectional GCM results, RnR identifies a dominant unidirectional pathway originating from the DMN.

Table 1: Comparison of Top 5 Causal Connections Identified by GCM (Temporal) and RnR (Structural) Methods

| GCM (Temporal Causality) | | | RnR (Structural Causality) | | |
|---|---|---|---|---|---|
| **Connection** | **Frequency** | **Interaction Type** | **Connection** | **Frequency** | **Interaction Type** |
| rFIC $\leftrightarrow$ rDLPFC | 58.1% | Salience $\leftrightarrow$ CEN | VMPFC $\rightarrow$ rFIC | 89.7% | DMN $\rightarrow$ Salience |
| VMPFC $\rightarrow$ PCC | 39.4% | DMN (internal) | rDLPFC $\rightarrow$ ACC | 67.3% | CEN $\rightarrow$ Salience |
| PCC $\rightarrow$ VMPFC | 39.0% | DMN (internal) | PCC $\rightarrow$ rFIC | 64.1% | DMN $\rightarrow$ Salience |
| VMPFC $\rightarrow$ rDLPFC | 38.4% | DMN $\rightarrow$ CEN | VMPFC $\rightarrow$ PCC | 62.6% | DMN (internal) |
| rDLPFC $\rightarrow$ VMPFC | 37.4% | CEN $\rightarrow$ DMN | VMPFC $\rightarrow$ rPPC | 59.1% | DMN $\rightarrow$ CEN |

## 5 CONCLUSIONS AND FUTURE DIRECTIONS

Undersampling is a critical yet often overlooked issue in the analysis of time series data, particularly in the context of fMRI. In this paper, we addressed the undersampling problem directly by incorporating its effects into the derivation of causal graphs using ASP. Our approach goes beyond traditional methods by explicitly accounting for the temporal disconnect between neural processes and fMRI sampling rates. By doing so, we not only identified the most probable causal graph but also provided an equivalence class of potential graphs, enabling a more nuanced understanding of the underlying causal structures.

Our results, validated on simulated fMRI data underscore the importance of addressing undersampling in neuroimaging studies. We demonstrated that our ASP-based method outperforms existing techniques, particularly in scenarios where undersampling distorts the true causal relationships between neural groups. Additionally, by comparing our approach to other algorithms, including those studied by Sanchez-Romero et al. (2019b), we highlighted that while methods like FASK show potential, they still fall short in fully capturing the complexities introduced by undersampling. Finally, in application to empirical human fMRI data lacking ground truth, we observe that incorporating undersampling-aware constraints via ASP yields markedly different effective connectivity estimates. This divergence suggests that rigorously accounting for undersampling effects reveals previously obscured network dynamics, thereby offering novel mechanistic insights and opening new avenues for future investigation.

The undersampling issue is not merely a technical challenge but a fundamental barrier to accurate causal inference in neuroscience. Ignoring it risks drawing erroneous conclusions about brain function and connectivity. Our work represents a significant advancement in this area, providing a robust and scalable solution that can improve the accuracy of causal inference in the presence of undersampling.

As the scientific community continues to explore the neural mechanisms underlying cognition and behavior, it is imperative that the undersampling problem is given the attention it deserves. Addressing this issue will be crucial in ensuring that the inferences drawn from fMRI data truly reflect the underlying neural dynamics, leading to more reliable and meaningful insights into brain function.

One challenge for solving the optimization problem with ASP is ensuring a reasonable initial estimated graph $\mathcal{H}$. The estimation errors at the measurement time-scale may inflate the estimation errors at the causal timescale. However, simply selecting the estimator that minimizes the errors in $\mathcal{H}$, as we have done in this paper, may not be the optimal strategy. Not all errors in $\mathcal{H}$ have the same effect on the quality of estimation and developing methods that consider that interplay is a promising future direction.

Further optimization of these approaches to enable work with larger graphs may open potential new domains where our methods may be applied. Although RnR is highly capable for working with reasonably sized graphs, extending the number of nodes by an order of magnitude could broaden the range of potential applications. This includes further practical application of the methods in the study of brain function via fMRI as well as other dynamic modalities.

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

# Appendix

## A  Hyperparameter Optimization for RnR Priorities

### A.1  Motivation and Methodology

Our RnR algorithm employs a weighted optimization framework where five priority parameters control the relative importance of different constraints in the Clingo answer set programming solver. These priorities, denoted as $\mathbf{p} = [p_1, p_2, p_3, p_4, p_5]$, govern the trade-offs between: (1) minimizing false positive directed edges ($p_1$), (2) minimizing false positive bidirected edges ($p_2$), (3) minimizing false negative directed edges ($p_3$), (4) minimizing false negative bidirected edges ($p_4$), and (5) matching the expected graph density ($p_5$). Each priority can take integer values from 1 (lowest) to 5 (highest), resulting in $5^5 = 3{,}125$ possible configurations[7]. In Equation 1, $C_o(G)$ corresponds to $p_5$, $C_b(G)$ corresponds to $p_2, p_4$, and $C_d(G)$ corresponds to $p_1, p_3$. Crucially, the aggregate parameters $\lambda_o$, $\lambda_b$, and $\lambda_d$ in Equations $2 - 4$, derived from these components do not function as continuous coefficients typical in gradient-based learning. Instead, they represent distinct optimization priority levels that enforce a strict lexicographic ordering within the ASP solver. Consequently, the systematic tuning of $\mathbf{p}$ establishes the hierarchical relationship between these terms—determining, for instance, whether satisfying biological density constraints ($\lambda_o$) takes precedence over resolving edge orientation ($\lambda_d$). Thus, this optimization process defines the solver's structural decision logic rather than adjusting scalar penalty magnitudes.

To systematically identify the optimal priority configuration for causal direction recovery, we conducted an exhaustive hyperparameter search using high-performance computing resources. The search was designed to maximize orientation F1 score, which measures the accuracy of recovered edge directions—the most critical metric for causal inference. We employed a SLURM-based parallel computing framework where each of the 3,125 priority configurations was evaluated independently on a computing cluster. For each configuration, we generated data from a known ground truth causal graph (network 3 of simple networks from Sanchez-Romero et al. (2019b)) and evaluated performance across five independent batches to ensure statistical robustness. The PCMCI algorithm with partial correlation tests was used for initial graph estimation, followed by RnR. Performance metrics were computed by comparing the recovered causal-timescale graph against the ground truth.

### A.2  Experimental Results

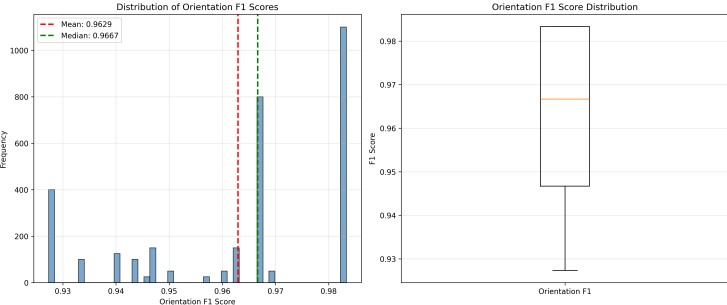

Figure 8: Distribution of orientation F1 scores across all 3,125 tested priority configurations. The distribution exhibits bimodal behavior with a primary mode near 0.98 and a secondary mode near 0.93. Red dashed line: mean (0.9629); green dashed line: median (0.9667).

The exhaustive search revealed substantial variation in performance across priority configurations (Figure 8). Orientation F1 scores ranged from 0.927 to 0.983, with a mean of $0.963 \pm 0.020$ (mean $\pm$ std) and median of 0.967. The distribution exhibited a bimodal pattern, with a dominant mode

---

[7]In practice, possible configurations are much less, since for example $[1, 1, 1, 1, 2]$ and $[1, 1, 1, 1, 5]$ are practically the same. Our hyperparameter tuning experiments confirms this finding as well.

of high-performing configurations (F1 $\approx$ 0.98) and a secondary cluster of lower-performing configurations (F1 $\approx$ 0.93), suggesting distinct regions of the priority space correspond to qualitatively different optimization behaviors.

The optimal priority configuration was identified as $\mathbf{p}^* = [1, 2, 1, 2, 3]$, achieving an orientation F1 score of 0.983. This configuration prioritizes maintaining correct graph density ($p_5 = 3$) above all other constraints, while assigning moderate importance to matching bidirected edges ($p_2, p_4 = 2$) and lower importance to directed edge constraints ($p_1, p_3 = 1$).

The systematic hyperparameter optimization yielded a 2.1% improvement in orientation F1 score over the baseline configuration (from 0.963 to 0.983), corresponding to a 56% reduction in orientation errors. This improvement, while seemingly modest, is significant given the already high baseline performance and the critical nature of causal direction accuracy in downstream causal inference tasks.

## B  Hyperparameter Optimization for Solution Selection Threshold

### B.1  Motivation and Methodology

Our RnR algorithm typically returns multiple candidate solutions with associated optimization costs. Rather than selecting a single minimum-cost solution, averaging across multiple near-optimal solutions can improve robustness by mitigating the impact of optimization noise and local minima. The critical question becomes: which solutions should be included in this ensemble?

We introduce a cost-based selection threshold, $\delta$, defined as a percentage of the minimum solution cost. Specifically, given a set of candidate solutions $\mathcal{S} = \{(G_i, c_i)\}$ where $G_i$ represents a causal graph and $c_i$ its optimization cost, we select the subset:

$$\mathcal{S}_\delta = \{(G_i, c_i) \in \mathcal{S} : c_i \leq c_{\min} + \delta \cdot c_{\min}\}$$

where $c_{\min} = \min_i c_i$. The parameter $\delta$ thus controls the solution selection bandwidth: $\delta = 0$ selects only the minimum-cost solution, while larger $\delta$ values include progressively more solutions with higher costs. The final causal graph estimate is obtained by averaging orientation F1 scores across all solutions in $\mathcal{S}_\delta$.

To identify the optimal threshold, we conducted a systematic hyperparameter search testing 21 values of $\delta \in [0\%, 200\%]$ in 10% increments. Each threshold was evaluated on network 3 with undersampling rate 3 across five independent batches for statistical robustness. For each batch, PCMCI with partial correlation tests generated an initial graph estimate. The orientation F1 score—measuring edge direction accuracy—was computed for all solutions within the $\delta$ threshold and averaged to obtain the final performance metric.

### B.2  Experimental Results

The hyperparameter search revealed a clear optimal threshold at $\delta^* = 90\%$ of the minimum cost, achieving an orientation F1 score of 0.7996 (Figure 9, top-left). Performance exhibited moderate sensitivity to the threshold choice, with F1 scores ranging from 0.7940 to 0.7996 across all tested values—a variation of approximately 0.7%. The optimization landscape showed a distinct peak at $\delta^* = 90\%$, with performance degrading for both smaller thresholds ($\delta < 90\%$) and larger thresholds ($\delta > 100\%$).

Precision and recall metrics remained relatively stable across the threshold range (Figure 9, top-right), with precision hovering around 0.790 and recall around 0.802. This stability suggests that the threshold primarily affects which near-optimal solutions are averaged rather than fundamentally altering the precision-recall trade-off. The slight decline in precision at very high thresholds ($\delta > 150\%$) indicates that including solutions with costs substantially above the minimum begins to introduce lower-quality predictions.

The number of solutions selected increased modestly with threshold, from approximately 103 solutions at $\delta = 0\%$ to 105 solutions at $\delta = 200\%$ (Figure 9, bottom-left). The near-horizontal trend indicates that the majority of DRASL solutions cluster tightly around the minimum cost, with few

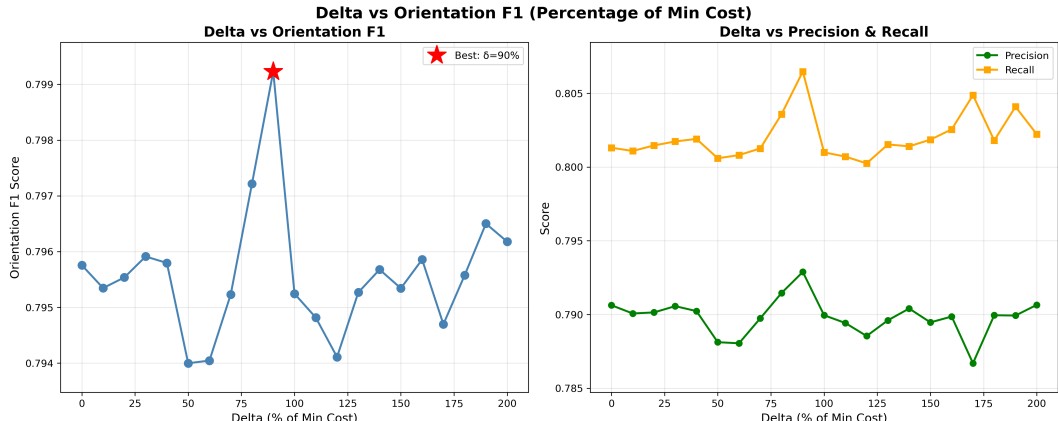

Figure 9: Hyperparameter optimization for solution selection threshold $\delta$. **Left:** Orientation F1 score versus $\delta$ (percentage of minimum cost). Red star indicates optimal threshold at $\delta^* = 90\%$, achieving F1 = 0.7996. **Right:** Precision and recall trends show stability across thresholds with slight precision decrease at higher $\delta$ values.

solutions falling in the intermediate cost range. The horizontal dashed line at 126 solutions represents the total solution set size, showing that even at $\delta = 200\%$, only 83% of solutions are selected. This distribution suggests that DRASL produces a concentrated set of high-quality solutions near the optimum and a separate tail of substantially suboptimal solutions.

### B.3 ANALYSIS AND IMPLICATIONS

The optimal threshold $\delta^* = 90\%$ represents a balance between solution quality and ensemble diversity. At this threshold, solutions with costs up to 1.9 times the minimum are included, selecting approximately 105 out of 126 available solutions. This relatively broad inclusion criterion suggests that DRASL's cost function, while effective at guiding optimization, may not perfectly correlate with orientation accuracy—solutions with moderately higher costs can still contribute positively to the ensemble average.

The percentage-based formulation of $\delta$ provides generalizability across different networks and problem scales. Unlike an absolute cost threshold, which would require recalibration for each experimental configuration, the percentage-based approach adapts automatically to the characteristic cost scale of each problem. This scale-invariance is particularly valuable for automated analysis pipelines processing heterogeneous datasets.

Comparing solution selection strategies, the ensemble approach with $\delta^* = 90\%$ (F1 = 0.7996) offers a modest but consistent improvement over single-solution selection ($\delta = 0\%$, F1 = 0.7958), corresponding to a 9.5% reduction in orientation errors. This improvement comes at minimal computational cost, as all candidate solutions are already generated during DRASL optimization. The robustness gained from averaging 105 solutions may prove particularly valuable in scenarios with high optimization noise or ambiguous causal structures.

The relatively flat performance plateau for $\delta \in [70\%, 110\%]$ (F1 variation $< 0.2\%$) indicates that the optimization is robust to moderate threshold misspecification. This tolerance provides practical flexibility in deployment scenarios where precise threshold tuning may be infeasible. All subsequent analyses employ the optimized solution selection threshold $\delta^* = 90\%$ of the minimum cost.

## C RNR ALGORITHM PSEUDOCODE

To improve reproducibility and address clarity concerns, we provide an explicit algorithmic description of our method. This section serves as a high-level implementation guide and complements the formal optimization framework presented earlier. Figure 2 shows the overall workflow of our method. We define the cost function as:

$$C(G) = \lambda_d \left( \sum_{e \in H_{dir}} w_e^+ \cdot \mathbb{I}[e \notin G] + \sum_{e \notin H_{dir}} w_e^- \cdot \mathbb{I}[e \in G_{dir}] \right)$$
$$+ \quad \lambda_b \left( \sum_{e \in H_{bi}} w_e^+ \cdot \mathbb{I}[e \notin G] + \sum_{e \notin H_{bi}} w_e^- \cdot \mathbb{I}[e \in G_{bi}] \right) \tag{5}$$
$$+ \quad \lambda_o \cdot \left| \sum_{e \in \mathcal{E}} \mathbb{I}[e \in G] - D^* \right|,$$

where $H_{dir}$ and $H_{bi}$ denote the sets of directed and bidirected edges in the measurement graph $H$, respectively. $G_{dir}$ and $G_{bi}$ denote the directed and bidirected edges in the candidate graph. $D^*$ represents the expected target edge count (density). The weights $\lambda_d, \lambda_b, \lambda_o$ correspond to the hierarchical priorities defined in Section 3.3.

The weights $w_e^+$ and $w_e^-$ in the cost function (Equation 5) critically influence which edges the solver prefers to keep or drop. We devise an adaptive weighting scheme based on the intuition that not all edges in the initial graph $\mathcal{H}$ are equally reliable, and similarly not all potential missing edges are equally implausible. Specifically, we leverage the strength of connection evidence (e.g., correlation or a statistical score) for each edge to set these weights: For each edge $e$ that is present in the initial graph $\mathcal{H}$ (meaning the first-order method or data proposed this connection), we assign $w_e^+$ inversely proportional to the confidence or strength of $e$. For example, if $e$ corresponds to a high correlation or a strong Granger causality score between two regions, we give it a high weight—the ASP solver will incur a large cost for removing this edge, so it will likely keep it unless necessary. Conversely, if $e$ was only weakly supported (perhaps a borderline significant link), we set $w_e^+$ lower, indicating that dropping this edge is not heavily penalized. This adaptivity encodes our uncertainty: strong edges should be trusted more (harder to delete), weak edges can be pruned if they conflict with other constraints.

For each potential edge $e$ that is absent in $\mathcal{H}$ (the initial method found no link), we set $w_e^-$ to a maximum penalty by default, reflecting a strong bias against adding completely new edges unless absolutely necessary. In other words, we assume $\mathcal{H}$ is mostly correct in saying those edges are absent, so the solver will avoid introducing a new connection $e$ because it would incur a high cost. However, there is one exception: if we have domain knowledge or secondary evidence that a certain missing edge might actually exist (e.g., known anatomical connection between those regions), we could lower $w_e^-$ for that edge. In general, though, $w_e^-$ is large to favor sparse solutions that do not introduce unsupported edges.

This weighting strategy was implemented by extracting pairwise correlation magnitudes from the time-series data for all region pairs. For instance, in our fMRI simulations, we computed the Pearson correlation for each pair of nodes; if an edge was present in $\mathcal{H}$, we mapped its correlation (or other score) to a weight in [0,1] range and then scaled it. Edges not in $\mathcal{H}$ were given a uniform high weight (effectively a large constant). The ASP solver thus solves a weighted min-change problem, where changes to highly credible edges are very costly, and changes to dubious edges are cheaper.

This adaptive weighting greatly improved the accuracy of the recovered graphs. It naturally handles the uncertainty in fMRI: for example, if two regions had a very low measured correlation yet some method erroneously connected them, RnR will likely remove that edge (low penalty to drop) in favor of satisfying structural constraints. On the other hand, if two regions are strongly correlated but the initial method missed the connection (say due to a too-conservative threshold), RnR will still be unlikely to add it because of the high default cost for new edges. However, if adding that edge is crucial to satisfy the DAG or density constraint, then the solver might do so, incurring the cost but yielding a more consistent graph. In summary, adaptive weights allow RnR to "decide" which edges to trust and which to doubt, rather than treating all edges uniformly.

## C.1 THEORETICAL GUARANTEES OF THE OPTIMIZATION PROCEDURE

While the presence of measurement noise and undersampling precludes a theoretical guarantee of statistical identifiability, ensuring the recovery of the unique ground-truth graph $\mathcal{G}^*$, our ASP-based

framework provides rigorous algorithmic guarantees regarding the optimization process. Unlike heuristic search methods (e.g., greedy hill-climbing) that may converge to local minima, the branch-and-bound optimization strategy employed by the solver guarantees global optimality for the defined cost function $C(G)$ Gebser et al. (2012). Furthermore, the retrieval of the candidate set is exhaustive: by leveraging the completeness of stable model semantics (Lifschitz (2002)), RnR guarantees the identification of the entire equivalence class of graphs satisfying the condition $C(G) \leq \Delta$, rather than a stochastic sample of high-scoring candidates. This ensures that the solution set captures the full extent of structural uncertainty compatible with the specified constraints.

---

**Algorithm 1** RnR Causal Graph Recovery Algorithm

---

**Require:** Initial graph $H$ (separated into directed $H_{dir}$ and bidirected $H_{bi}$ sets), edge weights $w^+, w^-$, target density $D^*$, ASP solver
**Ensure:** Final causal graph $G^*$ or set of plausible graphs $\{G_1, ..., G_k\}$
    **/* Encode ASP Optimization Problem */**
  1: Represent all candidate edges as ASP decision variables.
  2: Encode structural constraints:
      • Ensure no directed cycles between strongly connected components (DAG constraint).
      • Constrain solution space to biologically plausible density limits around $D^*$.
  3: Define hierarchical cost function (Eq. 5): $C(G) = \lambda_d C_{\text{dir}} + \lambda_b C_{\text{bi}} + \lambda_o C_{\text{density}}$
  4: If $H$ is from another method (meta-solver mode):
      • For any bidirectional link in $H_{bi}$, allow both $X \to Y$ and $Y \to X$ in the search space with low cost to resolve latent confounding.
    **/* Solve via ASP */**
  5: Use Clingo in `--opt-mode=optN` to retrieve all near-optimal solutions satisfying constraints.
  6: Collect output graphs $S = \{G_1, G_2, ..., G_k\}$.
    **/* Post-processing */**
  7: **if** $k > 1$ **then**
  8:    Identify edge agreement across solutions (robustness).
  9:    Optionally rank or filter graphs by validation data or consensus.
10: **end if**
11: Return $G^*$ as the best or consensus graph from $S$.

---

# D  BRIEF INTRODUCTION TO `clingo` AND ANSWER SET PROGRAMMING (ASP)

This section provides a brief technical overview of Answer Set Programming (ASP) and the solver `clingo`. ASP is a declarative problem-solving paradigm tailored for difficult combinatorial search and optimization problems. Unlike imperative programming, where the user specifies *how* to compute a solution, in ASP the user specifies *what* constitutes a problem and its valid solutions.

The tool `clingo` is a monolithic system that integrates two primary processes:

1. **Grounding (Gringo):** Instantiates a logical program containing variables into a variable-free (propositional) format.

2. **Solving (Clasp):** Performs conflict-driven nogood learning (CDNL) to find stable models (answer sets) that satisfy the grounded logic program.

A typical ASP encoding follows a "Generate-Define-Test" methodology. Below, we outline the core syntactic constructs of `clingo` using the classic *Graph Coloring Problem* as an illustrative example.

## D.1  FACTS: THE PROBLEM INSTANCE

Facts represent the unconditional truths of the specific problem instance. These are atoms (predicates) that are true at the start of the solving process. In a graph context, facts usually define the topology, such as nodes and edges.

Listing 1: Defining a graph instance and available colors.

```
1 node(1..4).           % Shorthand for node(1), node(2), etc.
2 edge(1,2). edge(2,3). edge(3,4). edge(4,1). edge(1,3).
3 color(red). color(blue). color(green).
```

In Listing 1, `node(1..4)` utilizes `clingo`'s range syntax to generate facts for nodes 1 through 4 efficiently.

## D.2 RULES: LOGICAL DERIVATION

Normal rules are used to derive new information based on existing facts or other derived atoms. A rule is structured as `Head :- Body.` and follows the logic: "If the Body is true, then the Head must be true." Variables in ASP are denoted by uppercase letters and are universally quantified.

For undirected graphs, we might want to ensure that if an edge exists from $X$ to $Y$, the system treats it as symmetric:

Listing 2: Deriving symmetric edges.

```
1 % If there is an edge from X to Y, imply an edge from Y to X
2 edge(Y, X) :- edge(X, Y).
```

During the grounding phase, `clingo` replaces variables $X$ and $Y$ with all valid constants from the domain (e.g., numbers 1 to 4) to create propositional clauses.

## D.3 CHOICE RULES: THE GENERATOR

To solve combinatorial problems, the solver must explore the search space by making choices. This is achieved via *choice rules* (or cardinality constraints). A choice rule allows the solver to select a subset of atoms to include in a candidate model.

The syntax `{ A } = N` asserts that exactly $N$ atoms from set $A$ must be true.

Listing 3: Generating color assignments.

```
1 % For every node N, choose exactly one color C from the defined colors.
2 1 { assign(N, C) : color(C) } 1 :- node(N).
```

In Listing 3, the construct `1 { ... } 1` enforces a strict constraint: for each specific node $N$, the solver must pick a minimum of 1 and a maximum of 1 color assignment. This generates the search space of all possible valid and invalid colorings.

## D.4 INTEGRITY CONSTRAINTS: THE TESTER

While choice rules generate the potential search space, *integrity constraints* prune this space by eliminating invalid solutions. An integrity constraint is a rule with an empty head, written as `:- Body.` It signifies: "It must not be the case that the Body is true."

If the body of an integrity constraint is satisfied by a candidate model, that model is discarded.

Listing 4: Eliminating invalid graph colorings.

```
1 % It is strictly forbidden for two connected nodes to share a color.
2 :- edge(X, Y), assign(X, C), assign(Y, C).
```

Listing 4 ensures that if two nodes $X$ and $Y$ share an edge, they cannot be assigned the same color $C$. If the solver attempts a model where connected nodes are both red, this constraint triggers, and the solver backtracks.

## D.5 SOLVING AND ANSWER SETS

When `clingo` executes the program combining the facts, generator, and constraints, it outputs *Answer Sets*. Each Answer Set is a minimal stable model representing a valid solution to the problem.

For the example above, a resulting answer set might look like:

```
assign(1,red) assign(2,blue) assign(3,green)
assign(4,blue) ...
```

If no model satisfies all constraints (e.g., trying to color a fully connected graph of 4 nodes with only 3 colors), `clingo` will return `UNSATISFIABLE`.

### D.6  ADVANCED FEATURES: OPTIMIZATION

`clingo` also supports optimization statements to find the "best" models among the valid ones. For example, if we assigned a cost to using specific colors, we could instruct the solver to minimize the total cost:

```
1  % Minimize the sum of Cost for all chosen assignments
2  #minimize { Cost,N : assign(N, C), cost(C, Cost) }.
```

This feature allows `clingo` to be used not just for satisfiability tasks, but for complex optimization problems common in machine learning pipelines and logic-based reasoning.

