# OpenReview forum: "Causal Graph Recovery in Neuroimaging through Answer Set Programming"
_ICLR.cc/2026/Conference — ICLR 2026 Conference Desk Rejected Submission_

### Official Review · Reviewer_sNXs · 2025-10-20

**Soundness:** 3
**Presentation:** 2
**Contribution:** 3
**Rating:** 6
**Confidence:** 2

**Summary:**

The paper introduces RnR, an ASP-based meta-solver that recovers more accurate causal graphs from undersampled fMRI data by enforcing neuroscientific and structural constraints. It extends prior Rate-Agnostic Structure Learning (RASL/sRASL) methods and bridges the gap between fast neural processes and slow fMRI measurements.

**Strengths:**

1. The paper takes undersampling into account when doing causal discovery and directly encodes the mathematical structure of undersampling (compressed paths and latent confounding) into the ASP formulation.

2. RnR retrieves all graphs within a small cost tolerance, effectively approximating the Markov equivalence class of causal structures compatible with data, which provides more robust solutions.

3. On simulated and real fMRI data, RnR improves F1-scores by an average of ≈12% over state-of-the-art baselines. It maintains high accuracy even when undersampling worsens, while competitors degrade sharply.

**Weaknesses:**

1. RnR does not discover causal edges directly from raw time series. It starts from an input graph generated by another causal discovery algorithm (Granger, PCMCI, FASK, etc.). If the initial graph is not accurate, the output of RnR may be affected.

2. RnR needs to solve a combinatoric problem using ASP. For graphs with a lot of variables, this optimization problem may be intractable.

**Questions:**

1. Can you add a complete pipeline to the overview of your method? It helps readers who are not familiar with the medical settings understand your paper better. For example, I know the input of RnR is the output of some causal discovery algorithm until section 3.5.

2. In practice, how do you decide the size of the tolerance?

3. How do you ensure acyclicity in ASP?

---

> ### Author Response · Authors · 2025-11-28
> **discussion about meta approach and combinatorial nature of method; pipeline figure added. appendix added for hyperparameter tuning $\Delta$**
>
> >not discover causal edges directly from raw time series
>
> Indeed, RnR is not a causal discovery algorithm in the full sense of the term - it does not work from the raw data samples to produce a causal graph - but it is a causal discovery enhancer, that augments other capable causal discovery algorithms with an ability to account for the unknown discrepancy between the causal and measurement timescales. We, however, view this as a strength, not a weakness. In the spirit of the general flexibility of our ASP approach, where additional constraints or prior knowledge may be easily encoded thus enhancing RnR by adapting it to the problem at hand, we view the ability to change the base discovery algorithm as a modularity advantage. Different data modalities and data domains may benefit from different causal discovery algorithms — in our case, for instance, PCMCI showed stronger performance — so an ability to select a causal solver suitable for the job is a plus. RnR will only work better if the base solver reduces the number of estimation errors.
>
> We agree with the reviewer that an interesting future work would be to tie the inherently combinatorial problem solved by RnR into a closed loop with the discovery algorithm.
> >needs to solve a combinatorial problem using ASP
>
> We appreciate the reviewer’s concern regarding computational tractability. However, the fundamental motivation for employing Answer Set Programming (ASP) is precisely its highly optimized capability to solve such combinatorial NP-hard problems in manageable time. As demonstrated in the sRASL paper (Abavisani et al., 2023), reformulating this specific causal discovery problem into an ASP framework improved solving time by approximately 3-4 orders of magnitude (1000-fold) compared to previous solvers. Consequently, this encoding renders previously intractable optimizations feasible, allowing the method to scale efficiently to graphs with over 100 nodes
> > complete pipeline figure
>
> We thank the reviewer for their insightful comment. We have created a pipeline figure and added to the newly uploaded version of the manuscript.
> >Decide the size of tolerance $\Delta$
>
>  We apologize for the oversight of not including the Appendix in the first submission. Appendix A&B details our hyper parameter tuning procedures for $\Delta$ and  $\lambda_{o}, \lambda_{b}, \lambda_{d}$.Appendix A and B confirm that our approach is highly stable and not overly sensitive to hyperparameter selection. Our analysis of all 3,125 priority configurations reveals that the method is robust to parameter changes, achieving a mean Orientation F1 score of 0.963 and a median of 0.967, which are only marginally lower than the optimal score of 0.983. This indicates that high performance is maintained across the vast majority of the parameter space rather than being confined to a narrow optimum. Similarly, the solution selection threshold (δ) exhibited remarkable stability, with performance varying by less than 0.7% across a wide range of values (0%–200%) and showing a distinct plateau of robustness between 70% and 110% where variation was under 0.2%.
> >How do you ensure acyclicity in ASP?
>
>  we are adopting the compressed representation of causal graphs following original RASL paper [ Plis et al. work Rate-Agnostic (Causal) Structure Learning NeurIPS2025], where cycles are allowed since every edge is from a past node to a future node. For example consider the graph with two node A,B at time points t, t+1, .. . An edge from $A_{t} \rightarrow B_{t+1}$ in the unrolled representation translates to $A \rightarrow B$ Proverbially, there is a one to one mapping between the two representations.  Therefore, by design and by definition, every edge that ASP find in the solution set is an edge from past to future. Consequently, we do not enforce strict acyclicity between individual nodes but this does not violate causal assumptions.

---

### Official Review · Reviewer_sgoz · 2025-10-28

**Soundness:** 4
**Presentation:** 4
**Contribution:** 4
**Rating:** 8
**Confidence:** 4

**Summary:**

This paper introduces Real-world noisy RASL (RnR), an extension of solver-based RASL that leverages Answer Set Programming (ASP) for causal graph recovery from undersampled fMRI time-series. By embedding domain-specific constraints and retrieving multiple near-optimal solutions, RnR explicitly models undersampling effects that distort effective connectivity inference. The paper demonstrates improved causal accuracy on simulated and empirical BOLD data, especially compared with traditional approaches (e.g., GIMME, MVAR, FASK, PCMCI). The method can also act as a “meta-solver,” refining graphs produced by other algorithms.

**Strengths:**

1.	Exceptional clarity and structure.
The exposition is unusually well-organized for an fMRI causal paper—each method component (density constraints, adaptive weighting, SCC-DAG decomposition, meta-solver function) is clearly explained and mathematically specified.
2.	Strong contextual grounding.
The introduction and background display excellent awareness of prior work in fMRI causal discovery—Granger, MVAR, FASK, PCMCI, RASL, and newer deep learning variants. The authors accurately summarize the challenge of undersampling and how it induces structural artifacts.
3.	Methodological innovation.
Extending sRASL to produce an equivalence class of near-optimal graphs via ASP optimization tolerance (Δ) is a genuine contribution  ￼. The addition of density and adaptive-weight constraints reflects good domain insight.
4.	Conceptual coherence.
The argument that causal inference from fMRI must explicitly incorporate undersampling is persuasive. Their use of a meta-solver to adjust outputs from existing algorithms is elegant and pragmatically valuable.
5.	Readable and reproducible.
The methods are described with sufficient procedural detail (e.g., Clingo code, cost equations, prioritization steps) to permit reproduction, and the empirical sections link cleanly to earlier definitions.

**Weaknesses:**

1.	Limited empirical scale.
Current experiments involve small networks (≤ 10 nodes in ASP runs, up to 50 nodes in discussion)  ￼. While understandable, it would help to show scaling behavior or runtime analysis beyond toy graphs.
2.	Quantitative evaluation scope.
Most comparisons are against the Sanchez-Romero synthetic dataset and a few standard algorithms. A more diverse benchmark (e.g., larger synthetic networks, real multi-subject fMRI) would better test generality.
3.	Lack of undersampled-data citations.
The paper could more fully engage with recent literature on learning from undersampled or irregularly sampled time-series beyond the fMRI context (e.g., algorithms for compressed or subsampled causal discovery).
4.	Open theoretical questions.
Although the ASP cost structure is clearly defined, the theoretical guarantees (e.g., completeness, correctness of near-optimal equivalence class) are not formally analyzed. A brief discussion of what is and isn’t provably ensured would strengthen confidence.
5.	Computational feasibility.
The paper notes ASP may slow for 20–50-node graphs. It would be valuable to include wall-time plots or solver statistics to show practical limits, especially since scalability is crucial for fMRI.

**Questions:**

1.	Scaling: What strategies or heuristics do you foresee to make RnR feasible for 100+-node parcellations common in fMRI (e.g., Schaefer or Glasser atlases)?
2.	Solver robustness: How sensitive are the retrieved equivalence classes to the choice of tolerance Δ in Equation (1)?
3.	Empirical realism: Have you tested RnR on real multi-subject fMRI datasets (e.g., HCP or resting-state data), and if so, how consistent are the equivalence classes across subjects?
4.	Connection to undersampling literature: Could you cite additional work addressing causal inference under irregular or sub-Nyquist sampling outside fMRI, to situate RnR in that broader methodological landscape?
5.	Meta-solver evaluation: When refining graphs from other methods (Figure 3), how is improvement quantified—do you compare F1, orientation precision, or overall structure distance?

---

> ### Author Response · Authors · 2025-11-28
> **Thank you for feedback. fMRI experiment added, hyperparameter and robustness studies added in appendix**
>
> > scaling behavior for larger graphs
>
> We thank the reviewer for this valuable insight and agree that demonstrating scaling behavior is a critical direction, which we have explicitly designated as a priority for our future work. Our primary focus in this study, however, was to bring the current frameworks "closer to reality" by rigorously addressing real-world noise and undersampling artifacts in standard ROI-level networks. Importantly, RnR builds upon the sRASL framework—which has been previously demonstrated to scale to graphs with over 100 nodes —and inherits this efficiency; indeed, in our current experiments, ASP solving was extremely fast, typically completing in under a few seconds per instance.
> >Quantitative evaluation scope
>
> Inspired by your feedback, we have added a dedicated section in our results applying RnR to real human brain fMRI data, specifically analyzing interactions between the Default Mode, Salience, and Central Executive networks. We benchmarked our findings against established literature and standard Granger Causality Mapping, demonstrating that our method effectively recovers known connectivity patterns while revealing novel structural hierarchies often obscured by undersampling in traditional temporal analyses.
> >Lacking some citations
>
> We appreciate the reviewers’ suggestion to contextualize RnR within the broader landscape of causal inference under sampling constraints. In response, we have expanded Section 2 to explicitly engage with literature on irregular and sub-Nyquist sampling beyond neuroimaging. We now incorporate discussions on Granger-causality adaptations for irregularly sampled physiological data (Bahadori & Liu, 2012; Heerah et al., 2021) and causal compressive sensing approaches in genomics (Emad & Milenkovic, 2014). This revision clarifies that RnR contributes to a wider family of methods addressing acquisition below the Nyquist rate, offering a specialized ASP-based solution for the specific hemodynamic distortions found in fMRI.
> >theoretical guarantees
>
> Theoretical Guarantees of the Optimization Procedure: While the presence of measurement noise and undersampling precludes a theoretical guarantee of statistical identifiability, ensuring the recovery of the unique ground-truth graph $G*$, our ASP-based framework provides rigorous algorithmic guarantees regarding the optimization process. Unlike heuristic search methods (e.g., greedy hill-climbing) that may converge to local minima, the branch-and-bound optimization strategy employed by the solver guarantees global optimality for the defined cost function $C(G)$ (Gebser et al., 2012). Furthermore, the retrieval of the candidate set is exhaustive: by leveraging the completeness of stable model semantics (Lifschitz, 2002), RnR guarantees the identification of the entire equivalence class of graphs satisfying the condition $C(G) \le  \Delta$ , rather than a stochastic sample of high-scoring candidates. This ensures that the solution set captures the full extent of structural uncertainty compatible with the specified constraints.
> >Solver robustness
>
> We apologize for the oversight of not including the Appendix in the first submission. Appendix A&B details our hyper parameter tuning procedures for $\Delta$ and  $\lambda_{o}, \lambda_{b}, \lambda_{d}$.Appendix A and B confirm that our approach is highly stable and not overly sensitive to hyperparameter selection. Our analysis of all 3,125 priority configurations reveals that the method is robust to parameter changes, achieving a mean Orientation F1 score of 0.963 and a median of 0.967, which are only marginally lower than the optimal score of 0.983. This indicates that high performance is maintained across the vast majority of the parameter space rather than being confined to a narrow optimum. Similarly, the solution selection threshold (δ) exhibited remarkable stability, with performance varying by less than 0.7% across a wide range of values (0%–200%) and showing a distinct plateau of robustness between 70% and 110% where variation was under 0.2%.
> > RnR on real multi-subject fMRI dataset
>
> We appreciate the reviewer’s emphasis on empirical realism. In response, we have added a new experimental section applying RnR to real multi-subject fMRI data using ICA-derived brain regions. Our analysis demonstrates high consistency in the recovered equivalence classes across subjects. To quantify this, we report the frequency of specific edge presence across the subject pool, revealing robust structural connections—such as the VMPFC → rFIC link found in nearly 90% of cases—that persist across individuals.
> > Meta-solver evaluation
>
> In our experiment we evaluate for Precision and Recall and F1 score derived from precision and recall for 1- Adjacency(getting the presence of connection correct) 2-Orientation(getting the direction of edge correct) Due to space limitation, we only include F1 score in our figures. (Figure 4,5,6),

---

### Official Review · Reviewer_o9GW · 2025-10-31

**Soundness:** 2
**Presentation:** 3
**Contribution:** 2
**Rating:** 2
**Confidence:** 3

**Summary:**

This paper addresses causal discovery in time series with undersampling, where not all timepoints are observed. This setup is heavily motivated by fMRI and causal discovery for neuroscience. The work builds on top of the sRASL framework, with the authors adding various heuristics to this method motivated by their observations and the nature of the settings. The method allows recovery of more graphs by searching graphs within a range rather than only the best score. They add a penalty for not following the expected density and propose not only to use a graph as input but also to incorporate known correlations. They also address how to treat bidirected edges. The findings are evaluated against the previous framework on some data, and then the comparison is made on synthetic and semi-synthetic data against other approaches with various undersampling rates.

**Strengths:**

* An idea to increase the robustness of the method by recovering multiple solutions
* An interesting idea to use domain knowledge (about density) as a constraint
* A very interesting point is made in Section 3.5, where the authors propose to integrate into the process of recovering a graph not only the output graph of some algorithm but also the uncertainty about the edge or knowledge about the strength of the relation. Such an approach seems like a step in a good direction, not to "waste" any information.
* Compared RnR (proposed) to sRASL on an edge deleting experiment.
* Compared to a broad range of methods.

**Weaknesses:**

* Subsections 3.2 and 3.4 both have a C(G) function, but they have different forms. Can you clarify which one you used?
* There is no discussion about tuning parameters of this method like δ or λ_d and λ_c.
* For all experiments, there is no hyperparameter tuning procedure described. However, since this is a heuristics method, I think it would be important to describe how robust they are, what is needed to tune them, and how general these hyperparameters are.
* Subsection 4.1 - How did you select the final graph? You write "and then select the final graph by an additional criterion (e.g., best fit on a validation dataset or expert judgment)" - so was the best fit chosen in this approach? Did you use all optimization steps described? If I understand correctly, the input to this experiment is a graph, so we only use the score component, meaning the modifications involving the adaptive weighting scheme for edges (Section 3.5) were not present?
* Subsection 4.2 - As far as I understand, the output of Sanchez-Romero's method can also be plugged as an input to sRASL. Why did you not directly compare RnR with this approach? Since these heuristics build on it, it should be compared. If this is not possible, please correct me and possibly explain why this is the case.
* Also, I think the appendix is missing.

**Questions:**

* Please look at some questions in the weakness section.
* In Section 3.6, you point out that methods like PC or GES will not output bidirected edges. But then you write it can lead to spurious edges like two-node cycles. However, I do not think that PC or GES will output any cyclic graph, because they output a CPDAG. Can you explain this?Also, in such a setting, is it not mandatory to use a method capable of discovery with confounders?
* Is there a reason why structural metrics like SHD are not used?

---

> ### Author Response · Authors · 2025-11-28
> **We thank the reviewer for their valuable and constructive feedback. We made major revisions inspired by their concerns. Appendix added**
>
> >  C(G) function discrepancy
>
> Formula in Section 3.2 is a more detailed version of formula in Section 3.4.e.g. The number of directed mismatches translates to $w^{-}$ and $w^{+}$ and the indicator function counts such mismatches. As the reviewer rightly points out the confusion this can cause, we kept the Section 3.4 in main text and moved a more detailed version of Section 3.2 to Appendix C. Section 3.5 was also related to section 3.2. Therefore we moved that to Appendix C as well.
> > no discussion about tuning parameters
>
> We apologize for the oversight of not including the Appendix in the first submission. Appendix A&B details our hyper parameter tuning procedures for $\Delta$ and  $\lambda_{o}, \lambda_{b}, \lambda_{d}$. We confirm that our approach is highly stable and not overly sensitive to hyperparameter selection. Our analysis of all 3,125 priority configurations reveals that the method is robust to parameter changes. This indicates that high performance is maintained across the vast majority of the parameter space rather than being confined to a narrow optimum. Similarly, the solution selection threshold (δ) exhibited remarkable stability, with performance varying by less than 0.7% across a wide range of values (0%–200%) and showing a distinct plateau of robustness between 70% and 110% where variation was under 0.2%.
> > How you select final graph
>
> In general, as output we provide a set of solutions that are closest to the true answer. One can stop there and seek help from an expert in the field (e.g. a neuroscientist that has knowledge of the brain connections) to ultimately find the graph that is closest to reality. However, we can also use the optimization cost of each solution to pick the solution with minimum cost as the best answer. As we have shown in Figure 1.B, there is a negative correlation between optimization cost and error, meaning if we choose the answer with least cost we are more likely to pick a better solution. But due to the uncertain nature of the problem, suggest picking the top N(e.g. Top 10) answers as the final answer set. As we have shown in Figure 1.B, the red dots are members of a solution set for one input and the top 10 answers with least cost (bottom left corner) are a very good and robust set of answers all with low error.
>
> As for Section 4.1, we did use all the optimization steps described in Section 3. However since this experiment is the simplest case designed to showcase improvements over the latest state-of-the-art method, we simply generate graphs. As opposed to other sub sections in section 4 where more complicated simulations are tested. In Section 4.1, since the estimated graph is not inferred from time series data using causal discovery methods (SVAR, PCMCI,...) the edges are all treated similarly with weight =1.
> > Why not directly apply RnR to Sanchez-Romero's
>
> Indeed our method is a meta method that can be applied on top of any causal discovery method including  Sanchez-Romero's (FASK). In Sec 4.2 we describe this experiment and Figure 3  shows the significant improvement we get over the original Sanchez-Romero's FASK method (green) Vs.  Sanchez-Romero's + RnR meta (purple)
> > cycles PC or GES
>
> Indeed PC will not output a cycle in the unrolled representation of causal graphs with time steps. However, we are adopting the compressed representation of causal graphs following original RASL paper [ Plis et al. Rate-Agnostic (Causal) Structure Learning NeurIPS2025], where cycles are allowed since every edge is from a past node to a future node. For example consider the graph with two node A,B at time points t, t+1, .. . An edge from $A_{t} \rightarrow B_{t+1}$ in the unrolled representation translates to $A \rightarrow B$. Proverbially, there is a one to one mapping between the two representations.
> > mandatory to use a method capable of discovery with confounders?
>
> It is not mandatory. However, our method produces much better answers with such methods that can produce bidirected edges where there is a hidden confounder. If a method cannot discover cofounders, as detailed in Sec 3.6 we can assign bidirected edges where there is a two cycle(unrolled version: $A_{t} \rightarrow B_{t+1} , B_{t} \rightarrow A_{t+1} \equiv A \rightarrow B , B \rightarrow A$ ) then assign directed edge probabilistically.
> > why not SHD
>
> SHD is only well defined if you have a graph with only undirected edges, or directed edges. That is, we only consider a Bayesian network, or a causal DAG as candidates. If there are more than one type of edge within the network, then SHD can be called on a sub-graph of that edge type. For example, as no systematic method exists to compare PAGs containing circular edges, we instead evaluate the count of circle edges or the SHD of the directed, undirected, and bidirected subgraphs. We did use Omission and Commission for the simple edge breaking experiment (Figure 2), but used more commonly used F1 score for the rest of the experiment.

---

### Official Review · Reviewer_oBdh · 2025-11-01

**Soundness:** 2
**Presentation:** 3
**Contribution:** 2
**Rating:** 4
**Confidence:** 3

**Summary:**

I liked this paper. Undersampling happens in many cases, and we know our current techniques fail to recover the effective connectivity properly in undersampled settings. I also liked the clear and step-by-step presentation of the strength of the methodology proposed by the authors. That said, I am not sure if I am convinced that the method solves the issues we have. In what follows, I bring the concerns in my mind and I am happy to change the score I give here if the authors bring evidence that the method can address those indeed.

**Strengths:**

A quite clear question, clear method, and clear assessment

**Weaknesses:**

1. The authors claim that the method works for neuroimaging data, but they only test for fMRI. Does this method extend to other modalities e.g., fNIRS?

2. What happens if we remove the constraints of 10-30% density? If the method fails to converge, are we truly finding a causal structure or do we have some level of circularity?

3. When does the method fail completely? I suspect if the u (delay) is varied and long, the method would completely fail. This is important because we know the undersampling does not only happen because of the BOLD dynamics, but also because of the feedback loops in the brain. Could the others test the limits of the method theoretically or at least empirically? This could be possible for instance, when we simulate an arbitrary rich high-dimensional dynamical system with local linearity (e.g., piece-wise linear RNNs) and see if the method proposed here can indeed deliver the effective connectivity for each basin of attraction?

4. What happens to multistable systems? after all, BOLD itself is multistable and connectivity changes in time.

**Questions:**

see above.

---

> ### Author Response · Authors · 2025-11-28
> **confirming the method's applicability to fNIRS, clarifying that density constraints are optional priors that prune the solution space, and defining the method's scope regarding stationary structural recovery versus dynamic state changes.**
>
> >  method extend to other modalities?
>
> We thank the reviewer for raising this important question about the broader applicability of RnR. Yes. RnR is suitable for fNIRS because, like fMRI, it relies on slow neurovascular coupling that inherently undersamples fast neural activity. The method applies to any system where observed time series evolve slower than the underlying causal interactions. While electrophysiological modalities  capture activity at native scales and do not require this specific correction, integrating RnR with them to reconcile hemodynamic and electrical timescales is a promising future direction.
>
> >What happens if we remove the constraints of 10-30% density?
>
> The density constraint serves as a biologically informed prior rather than a mathematical requirement for convergence; removing it does not cause the method to fail, but simply results in a larger Markov equivalence class of solutions. While the solver remains capable of correctly recovering the complete set of statistically plausible causal structures, eliminating this constraint removes a valuable mechanism for pruning biologically unrealistic candidates from that broadened solution set.
>
> Regarding the concern about circularity, the density constraint operates independently of the structural validity of the graph. We explicitly prevent circularity by hard-coding constraints within our Answer Set Programming formulation to prohibit directed edges to past nodes. Compressed graph representation is a one-to-one mapping between the compressed model and the unrolled causal graph,and every edge in the compressed graph representation is an edge from past to future. Therefore, if for example there is a cycle between nodes A and B, it means in one time step node A affects node B and in the next time step, node B affects node A. Hence, causality stands ensuring that the recovered solution maps correctly to a valid temporal structure without introducing logical circularity.
>
> > method fail completely?  ...
>
> Regarding the limits of the method, our approach inherits the theoretical guarantees of the Rate-Agnostic Structure Learning (RASL) framework, meaning it is designed to solve for the causal structure without requiring prior knowledge of the specific undersampling rate u. However, you are correct that failure modes exist: the method hits an information-theoretic hard limit when u becomes large enough that the 'compressed' graph representation becomes fully connected (saturated). At that point, no algorithm can distinguish the true structure because the aliasing has destroyed the conditional independencies required for recovery. In the context of fMRI, however, the biological hemodynamic response acts as a natural low-pass filter; this limits the effective u to a range where our ASP solver can still successfully disambiguate the feedback loops (modeled via Strongly Connected Components ) from the undersampling artifacts.
> With respect to the suggested experiment on piecewise linear RNNs, we agree that this is a fascinating direction for future work in state-dependent or time-varying connectivity. However, the current RnR framework—like the standard effective connectivity methods we benchmark against (e.g., MVGC, GIMME)—assumes stationarity over the observed window to recover a single, robust underlying causal graph. Testing effective connectivity per 'basin of attraction' implies a regime-switching model where the causal structure changes based on the state space, which is a distinct problem class from the global structural recovery we address here. We believe our current extensive validations on varying densities and 'edge-breaking' scenarios provide the necessary empirical evidence for the method's robustness within the scope of stationary causal discovery.
>
> > multistable systems?
>
> This is an important point, as we agree that BOLD signals reflect a brain that is inherently dynamic and likely multistable. However, it is crucial to distinguish between the multistability of the signal dynamics (the activity traversing the network) and the stability of the causal structure (the effective connectivity) that constrains those dynamics. Our current implementation of RnR—like the standard methods we benchmark against, such as MVGC and GIMME—assumes stationarity over the observation window to recover the underlying structural 'backbone' that permits these dynamic states to exist.
> If applied to a system where the connectivity itself is rapidly changing, our method would recover the aggregate or most persistent causal pathways that explain the data across the entire time series. While resolving state-dependent connectivity changes is a vital goal for the field, one must first be able to robustly recover structure under undersampled conditions in a static regime. We view extending RnR to a time-varying or windowed framework to capture these shifts as the logical next step, building directly on the solver-based foundation established in this work

---

### Author Response · Authors · 2025-11-28
**Summery of changes to revision PDF**

We want to sincerely thank all the reviewers for their invaluable and insightful feedback. We made the following edits:

1. Pipeline figure of method added (New Figure 2)
2. New missing citations added to Section 2
3. Section 3 is re-written. The more bulky and detailed version of optimization formula is moved to Appendix C, along with old Section 3.5
4. New experiment was added (Section 4.5) that applies out method to real fMRI data of human brain. Brief discussion is also included.
6. Appendix A is added. Details about hyperparameter tuning of  $\lambda_{o}, \lambda_{b}, \lambda_{d}$
7. Appendix B is added. Details about hyperparameter tuning of  $\Delta$
8. More detailed version of old Equation 1 is added to Appendix C.
9. Overview algorithm of RnR is added to Appendix C.
10. A brief introduction to Clingo and Answer Set Programming is added to complement reader who are not familiar with ASP.

---

### Author Response · Authors · 2025-12-02
**Final Remarks and Brief Summary**

Dear Area Chair,

Given the score reversion, we wanted to extend our gratitude to reviewers constructive feedbacks, and to provide a brief summary below to assist your assessment.

**New Real-World Validation (Section 4.5):**

We applied RnR to real human fMRI data, replicating established findings while uncovering novel structural hierarchies. This directly answers **oBdh**’s utility concerns and **sgoz**’s request for empirical realism. Additionally, by reporting connection frequencies (e.g., 89.7%), we demonstrated the practical graph selection criteria requested by **o9GW**.

**Comprehensive Resolution of *o9GW*’s Concerns:**

We paid particular attention to the detailed technical feedback from **o9GW**, ensuring that every concern—from hyperparameter robustness (now in Appendices A & B) to mathematical notation consistency (Section 3 and Appendix C)—has been fully addressed and resolved in this revision.

**Methodological Clarity & Scope:**

We resolved **sNXs**’s request for a clearer overview by adding a complete pipeline visualization (Figure 2). We also clarified the method's applicability to other modalities (e.g., fNIRS) for **oBdh** and expanded Section 2 to include broader citations on irregular sampling (e.g., genomics) as suggested by **sgoz**.

We believe the paper's contribution is strong, the concerns are addressed, and we respectfully ask that the rebuttals and revised manuscript be weighed alongside the frozen scores.

Thank you for your time.

Sincerely,

The Authors

---

### Note · Program_Chairs · 2026-01-17
**Submission Desk Rejected by Program Chairs**

The following references in this submission do not refer to real documents and/or have major errors in bibliographic information:

 Ruben Sanchez-Romero, Michael W Cole, and David M Schnyer. Estimating effective connectivity by jointly modeling neural dynamics and structural constraints. NeuroImage, 200:243-256, 2019a.
Mahdi Salehi, Hamid R Karimi, Soroush Honari, and Axel P Hill. Calltif: Whole-brain causal discovery using fmri data. In Proceedings of the 30th International Joint Conference on Artificial Intelligence (IJCAI), pp. 2078-2084, 2021.